Tailored modulation of S100A1 and RASSF8 expression by butanediamide augments healing of rotator cuff tears

Lin Yuan 1
Guo Ruipeng 1
R Geng 2
Xu Bin brdm4896406284@163.com 1
1 Department of Orthopedics, The First Affiliated Hospital of Anhui Medical University , Hefei , Anhui Province , China
2 Southeast University , Nanjing , China
Xu Yuzhen
Electronic publication date: 2023 Aug 14
Publication date: 2023
Volume: 11
Electronic Location ID: e15791
Received 2022 Dec 15; Accepted 2023 Jul 4
Copyright: ©2023 Lin et al.
Copyright year: 2023
Copyright holder: Lin et al.
License: This is an open access article distributed under the terms of the Creative Commons Attribution License, which permits unrestricted use, distribution, reproduction and adaptation in any medium and for any purpose provided that it is properly attributed. For attribution, the original author(s), title, publication source (PeerJ) and either DOI or URL of the article must be cited.
License URL: https://creativecommons.org/licenses/by/4.0/

Keywords: Rotator cuff tear (rct), S100A1, RASSF8, Zinc compound, TDSC

Funding: The authors received no funding for this work.

==============================
Objectives

This investigation sought to elucidate promising treatment modalities for rotator cuff tears (RCTs) by delving into the molecular machinations instigating the affliction. The focus was on differentially expressed genes (DEGs) linked to RCTs, and the exploration of their roles and operative pathways.

Methods

DEGs were discerned from GEO datasets, followed by the establishment of a protein-protein interaction (PPI) network. Subsequently, the network’s core genes were determined employing a Venn diagram. Enrichment analysis facilitated the unveiling of the biological roles and signal transduction pathways of these pivotal genes, thus shedding light on molecular strategies for RCT-targeted treatment. The Discovery Studio 2019 software was employed to sift through FDA-sanctioned drugs targeting these essential proteins. Moreover, the efficaciousness of these FDA-endorsed drugs vis-à-vis RCTs was corroborated by the construction of an in vivo animal model of the injury and the in vitro cultivation of tendon-derived stem cells.

Results

Bioinformatics outcomes revealed a significant overexpression of S100A1 and RASSF8 in RCT patients. The FDA drug repository indicated that Butanediamide has a selective affinity for S100A1 and RASSF8. Subsequent in vivo and in vitro experimentation demonstrated that Butanediamide could suppress S100A1 expression and bolster TDSC proliferation, thereby facilitating RCT healing.

Conclusions

S100A1 and RASSF8 are pivotal genes implicated in RCTs, and their roles have been elucidated. The FDA-approved compound, Butanediamide, may represent a prospective therapeutic agent for RCTs by targeting S100A1 and RASSF8, respectively.

Introduction

Rotator cuff tear (RCTs) is a common musculoskeletal disorder that becomes more frequent with age, with a prevalence of ultrasound-defined complete RCTs in the general population of 22.1% and increasing to 36.6% in those aged 80 years and older (Whittle & Buchbinder, 2015). Extrinsic risk factors of RCTs include occupation and exercise, while intrinsic risk factors consist of age, obesity, smoking, diabetes, genetic factors, and anatomical factors (Titchener et al., 2014). Patients with RCTs typically experience pain in the upper arm next to the deltoid muscle, which worsens at night or when lying on the affected side and is accompanied by impaired movement of the shoulder joint (Van der Windt et al., 1995). Most patients require arthroscopic repair treatment (Sims & Petsche, 2020; Cederqvist et al., 2021), which may be complicated by retear, shoulder stiffness, postoperative infections, suture anchor-related complications, deep vein thrombosis, and pulmonary embolism (Barth et al., 2017; Huberty et al., 2009; Vopat et al., 2016). The primary goal of rotator cuff repair surgery is to restore the tendon unit by creating an intact, tension-free repair structure to optimize tendon-to-bone healing conditions (Marrero, Nelman & Nottage, 2011). Currently, double-row suture bridge repair surgery has a higher healing rate and better clinical outcomes (Sheean, Hartzler & Burkhart, 2019). However, the specific molecular mechanism underlying rotator cuff repair remains unclear (Marrero, Nelman & Nottage, 2011).

The FDA-approved drugs is a list of drugs that have been approved by the Food and Drug Administration in the United States. The list includes various drug information, such as molecular structure, chemical identification, pharmacological effects, etc. Using the FDA-approved drugs can quickly screen for candidate drugs with potential therapeutic effects for specific diseases, and these drugs have undergone extensive testing, including clinical trials. Therefore, using the FDA-approved drugs for drug screening can save time and costs while also improving the safety and effectiveness of candidate drugs (Chen et al., 2016).

We analyzed differentially expressed genes (DEGs) associated with RCTs from the GEO database and constructed a protein-protein interaction (PPI) network to identify key proteins. Next, we performed a functional enrichment analysis to explore the molecule mechanism. The LibDock module was used for docking, and several FDA-approved compounds showed strong binding affinity for the core proteins S100A1 and RASSF8. Then, Chem3D software was used to create three-dimensional (3D) structures. In vitro experiments confirmed that S100A1 could suppress the proliferation of tendon-derived stem cells (TDSCs). In addition, through animal experiments, it was found that Butanediamide can significantly inhibit the expression of S100A1 and RASSF8, and promote healing of rotator cuff tears.

In conclusion, our study offers new understanding of the molecular mechanisms behind RCTs repair and proposes that FDA-approved drugs might have potential therapeutic benefits in RCTs treatment. The results of this research could serve as a theoretical basis for creating innovative approaches for the treatment of RCTs.

Methods

Identification of RCTs targets

Gene expression microarrays were searched using the keyword ‘RCTs’ in the GEO database of the National Center for Biotechnology Information (NCBI) with species selected as humans. The screened gene expression microarray data were normalized, and differentially expressed genes(DEGs) were identified based on p-values <0.05 and —log2 fold change (FC)—≥ 0.50. Heat maps and volcano maps were generated using the R package to demonstrate the DEGs associated with ‘rotator cuff injury’/‘RCTs’.

Establishment of protein to protein interaction network

The RCTs targets from different datasets were intersected using R (https://www.r-project.org/) and Perl software, and the Venn diagram was constructed using Venny website (http://bioinfogp.cnb.csic.es/tools/venny/index.html) to construct a Venn diagram. The intersected DEGs were imported into the STRING database (http://string-db.org/cgi/input.pl; version 11.0) with the following parameters: species, Homo sapiens; medium confidence; hide unconnected nodes. The protein-protein interaction(PPI) network was visualized using Cytoscape software(version 3.7.2), and the top 20 core proteins were identified based on their connection degree.

Gene Ontology (GO) and Kyoto Encyclopedia of Genes and Genomes (KEGG) enrichment analyses

The RCTs-DEGs and the intersected-DEGs were analyzed for GO and KEGG pathway enrichment analyses using the ‘clusterProfilerGO.R’ package in R and the Perl software, as with previous research (Wu et al., 2021; Sun et al., 2022; Luo et al., 2022). The core pathway enrichment degree was estimated based on the enrichment values to identify the potential biological functions and signalling pathways of the core targets of RCTs.

Screening of candidate drugs targeted to RCTs

Virtual screening was conducted using the LibDock module in Discovery Studio 2019. Six compounds with the best docking scores were selected for 2D and 3D presentation. The pdb files of the core protein structural domains were downloaded from the PDB database (http://www.rcsb.org/), and files of small molecules were downloaded from the ZINC15 database. Proteins were prepared by removing water of crystallization and other heteroatoms, hydrogenation, ionization, protonation, and energy minimization. The LibDock module was used to locate the docking site, and the binding energy and RMSD values of six compounds with the largest LibDockScore values were calculated. The PyMOL software was used to dehydrate and dephosphorylate the proteins, and the AutoDockTools (version 1.5.6) software was used to convert the six drug and core protein files from the pdb format to the pdbqt format. The AutoDock-Vina software was used to calculate molecular binding energy and visualize molecular docking results, with 3D and 2D presentation of ligand–receptor complexes, to evaluate the reliability of the predicted results (Goodsell DS & Olson, 1996; Kang et al., 2022; Lu et al., 2022; Zhang et al., 2022).

Experimental animals

Male Sprague-Dawley (SD) rats (age, 8 weeks) weighing approximately 200–250 g were purchased from Viton Lever. All animal procedures were approved by the Animal Research Ethics Committee of Southeast University (No.: 20201231016). Thiopental sodium (150 mg/kg) was used to euthanize all animals at the end of the experiment.

Experimental procedure

The experimental procedure and sample size were based on previous studies (Buchmann et al., 2011; Angeline et al., 2014; Buchmann et al., 2013; Bedi et al., 2010). Rats were randomly divided into drug and control groups to establish a rat model of rotator cuff reconstruction. The supraspinatus muscle was cut at the stop of the greater tuberosity of the humerus and sutured in place. To simulate acute rotator cuff injury, the supraspinatus tendon was dissected and a marrow tunnel was created from the medial forearm to the posterior lateral aspect of the greater tuberosity using a kerf needle. The tendon of the web muscle was then fixed in the bone tunnel, and shoulder joint mobility was examined to ensure good tension in the web muscle tendons. Non-absorbable sutures were used to close the surgical incision, and rats received iodophor disinfection and intramuscular injections of penicillin for infection prevention after recovering from anesthesia.

Drug administration

As described in previous studies (Yu et al., 2017; Zhang et al., 2018), rats in the drug group were fed normal chow with 100 mg/kg of butanediamide through daily oral administration from the first day after surgery. Butanediamide was synthesized by the Institute of Biology, Chinese Academy of Sciences. Rats in the control group were given saline at the same dose once daily. A total of 16 rats were sacrificed in the two groups, and samples were collected at 2 and 4 weeks after surgery to evaluate the effects of drugs on tendon-bone healing after rotator cuff reconstruction.

Isolation and culture of tendon-derived stem cells

Adult male rats were injected with chloral hydrate in their abdominal cavity, and the Achilles tendon was collected under sterile conditions. The tendon was cleaned of extra tissue and cut into small pieces of about 1 cubic millimeter. The tissue samples were placed in a solution with PBS containing type I collagenase (3 mg/mL per 100 mg of tissue) and neutral protease (4 mg/mL) and kept at 37 °C for 1 h. After incubation, the tissue was moved to a culture flask containing complete DMEM and grown in a 5% CO2 incubator with constant humidity for 24 h. The remaining tendon tissue was placed in a new culture flask and grown for about 10 days to obtain primary TDSCs. The cells were then transferred and grown for 2–3 generations before being used for further experiments. TDSCs cultured for 2–3 generations were transfected with 100-nM siRNA using Lipo 2000.

Protein extraction and western blotting

Transfected TDSCs were broken down in RIPA buffer (plus 1-mM protease inhibitor) at 4 °C to extract the total protein. The extracted protein was mixed with 5x loading buffer, denatured, and run equally in all wells on a 15% sodium dodecyl sulfate-polyacrylamide gel. The separated proteins were transferred onto a PVDF membrane after electrophoresis. The membrane was blocked for 2 h and incubated with primary antibodies against S100A1 (1:1000; Abcam, Cambridge, UK) and GAPDH (1:5000, CST, Danvers, MA, USA) overnight at 4 °C. The next day, the membrane was incubated with secondary antibodies for 1 h at room temperature, and protein bands were visualized using a Tanon system.

Cell proliferation assay

Cell proliferation was measured using a CCK-8 assay. The transfected cells (1 × 104 cells/mL) were placed in 96-well plates (100 µL/well). The CCK-8 assay was carried out every two days. In short, 10 µL of CCK-8 reagent was added to each well, and the plates were incubated at 37 °C for 2 h. The absorbance was measured at 550 nm using a microplate reader (Biorad, Hercules, CA, USA), and the results were plotted using GraphPad Prism software.

Statistical analysis

All experiments were carried out in triplicate, and the data were presented as mean ± standard deviation. Statistical analysis and graph creation were done using GraphPad Prism (version 7.0). Differences between groups were evaluated using one-way ANOVA and two-tailed t-tests. The western blotting and CCK-8 results were utilized to assess cell activity in different groups. Only p-values less than 0.05 were considered statistically significant.

Results

Differentially Expressed Genes (DEGs) associated with RCTs

To investigate the differentially expressed genes associated with rotator cuff tears, gene expression microarrays of RCTs from the GEO database (Accession numbers: GSE199484, GSE93661) were analyzed. The limma R package was used to create volcano plots and heat maps to display DEGs, which were based on —log2FC— ≥ 0.50 (more than 1.4-fold difference) and p-values of <0.05. In the GSE199484 dataset, 162 genes were downregulated and 95 genes were upregulated in the rotator cuff muscle belly tissues of patients with RCTs (Figs. 1A–1B). Similarly, in the GSE93661 dataset, 115 genes were downregulated and 84 genes were upregulated in the rotator cuff muscle tissues of patients with RCTs (Figs. 1C–1D).

Figure 1 Differentially expressed genes (DEGs) of rotator cuff tears (RCTs).

(A) Heat map of DEGs in GSE199484; (B) Volcano plot of DEGs in GSE199484; (C) Heat map of DEGs in GSE93661; (D) Volcano plot of DEGs in GSE93661 (Red represents upregulated genes and blue represents downregulated genes).

Protein–protein interaction network

A total of 257 and 199 target genes related to RCTs in the GSE199484 and GSE93661 datasets, respectively, were imported into the STRING database. The PPI network was created by hiding unconnected targets (Figs. 2A–2B). The top 20 core target proteins for disease-related co-expression were identified based on the Degree algorithm using the CytoHubba plugin (Fig. 2). The identified DEGs were matched and mapped with RCTs-intersected genes obtained from the GEO database, and duplicates were removed. Two target genes associated with RCTs were identified, S100A1 and RASSF8, which were increased in RCTs (Fig. 3).

Figure 2 PPI network of DEGs–RCTs.

(A) PPI network of DEGs in GSE199484; (B) PPI network of DEGs in GSE93661; (C) PPI network of the top 20 core proteins in GSE199484; (D) PPI network of the top 20 core proteins in GSE93661; (The darker the color, the higher the Degree value).

Figure 3 The Venn diagram shows the intersection of GSE199484 and GSE93661, with two common genes being S100A1 and RASSF8.

Enrichment analysis of RCTs-DEGs

In the GSE199484 dataset, GO analysis revealed that the target genes were enriched in biological functions such as muscle filament sliding, actin–myosin filament sliding, muscle system processes, muscle contraction, and actin-mediated cell contraction. KEGG analysis showed that these genes were enriched in adrenergic signaling in cardiomyocytes, cardiac muscle contraction, hypertrophic cardiomyopathy, dilated cardiomyopathy, and glucagon signaling (Figs. 4A–4B). In the GSE93661 dataset, GO analysis revealed that the DEGs were enriched in multi-organism cellular processes, positive regulation of response to wounds, histone modification, covalent chromatin modification, and regulation of the androgen receptor signaling pathway. KEGG analysis showed that they were mainly enriched in pathways associated with thyroid hormone signaling, lysine degradation, viral life cycle (HIV-1), transcriptional dysregulation in cancer, and central carbon metabolism in cancer (Figs. 4D–4E). The pathview package was used to visualize the signaling pathways associated with genes in the GSE199484 and GSE93661 datasets (Figs. 4C–4F).

Figure 4 GO and KEGG analyses of DEGs-RCTs in the GSE199484 and GSE93661 datasets.

(A) Bubble chart of GO enrichment analysis in GSE199484. (B) Bubble chart of KEGG enrichment analysis in GSE199484. (C) Calcium signaling pathway. (D) Bubble chart of GO enrichment analysis in GSE93661. (E) Bubble chart of KEGG enrichment analysis in GSE93661. (F) HIF-1 signaling pathway.

Enrichment analysis of core target genes

To further clarify the biological functions of the core target genes, S100A1 and RASSF8, the clusterProfiler GO R package and Perl software were used to perform GO and KEGG analysis. S100A1 and RASSF8 were found to be involved in various biological processes, including the positive regulation of voltage-gated calcium channel activity, cell–cell junction maintenance, and positive regulation of nitric oxide synthase activity (Fig. 5A). The genes were also found in cellular components such as the M band, A band, and sarcoplasmic reticulum (Fig. 5B). Molecular functions (GO-MF) include S100 protein binding, calcium-dependent protein binding, ATPase binding (Fig. 5C). The results are presented in a bar chart shown in Fig. 5D.

Figure 5 Enrichment analysis of the core genes.

(A) Bubble chart of biological processes. (B) Bubble chart of cellular components. (C) Bubble chart of molecular functions. (D) Histogram of GO functional analysis.

Drug screening of FDA database

The core proteins S100A1 and RASSF8 were acquired from the PDB database in pdb format and selected as receptor proteins. The ZINC15 database was used to access 1,615 molecules of FDA-approved drugs in mol2 format. A LibDockScore value greater than 100 was chosen to assess compounds with superior binding affinity and stability. LibDock module helped to identify 247 compounds that demonstrated good stability when bound to the core protein S100A1, and 374 compounds that demonstrated good stability when bound to the core protein RASSF8. Top 20 compounds as ranked by the LibDock scores are listed in Tables S1 and S2. Six compounds with optimal docking were chosen to measure their binding energy and RMSD values. The active pockets were located, and AutoDock Vina was used to compute the binding energy of ligands and receptors (Tables S3 and S4). The binding energy between S100A1 and ZINC000049841054, ZINC000085537014, ZINC000003941496, ZINC000003944422, ZINC000026985532 (butanediamide), and ZINC000049783788 was less than −5.0 kcal/mol. The RMSD values of the docking model formed by S100A1 and ZINC000049841054, ZINC000085537014, ZINC000003941496, ZINC000003944422, ZINC000026985532 (butanediamide), and ZINC000049783788 are also provided (Figs. 6 and 7).

Figure 6 Molecular docking model of S100A1.

(A–F) 3D Macro & Micro View, 2D Model of S100A1-ZINC000049841054, S100A1-ZINC000085537014. (G–L) 3D Macro & Micro View, 2D Model of S100A1-ZINC000003941496, S100A1-ZINC000003944422. (M–O) 3D Macro & Micro View, 2D Model of S100A1-ZINC000026985532 (Butanediamide). (P–R) 3D Macro & Micro View, 2D Model of S100A1-ZINC000049783788.

Figure 7 Molecular docking model of RASSF8.

(A–C) 3D Macro & Micro View, 2D Model of RASSF8-ZINC000028232750. (D–F) 3D Macro & Micro View, 2D Model of RASSF8-ZINC000003830635. (G–I) 3D Macro & Micro View, 2D Model of RASSF8-ZINC000095564694. (J–L) 3D Macro & Micro View, 2D Model of RASSF8-ZINC000085537014. (M–O) 3D Macro & Micro View, 2D Model of RASSF8-ZINC000036701290. (P–R): 3D Macro & Micro View, 2D Model of RASSF8-ZINC000072267023.

Inhibition of TDSCs proliferation by S100A1

Since tendon-derived stem cells (TDSCs) play a crucial role in the repair of RCTs, we investigated the effect of S100A1 on TDSC proliferation. Western blotting confirmed that si-S100A1 effectively reduced the expression of S100A1 (Figs. 8A–8B) (p = 0.036, t = 6.133, degrees of freedom = 4). Next, we performed a CCK-8 assay to assess the proliferation of TDSCs with si-S100A1 knockdown. The results showed that si-S100A1 promoted the proliferation of TDSCs (Fig. 8C) (72 h, p = 0.009, SD = 0.0812; 96 h, p = 0.011, SD = 0.1000).

Figure 8 S100A1 inhibited the proliferation of TDSCs.

(A) The results of the western blot showed that transfection with si-S100A1 effectively reduced the expression of S100A1. (B) Quantitative results of western blot. (C) The results of the CCK-8 assay revealed that transfection with si-S100A1 promoted TDSCs proliferation.

Discussion

Rotator cuff tears (RCTs) is a common movement system disorder that can cause shoulder pain and functional impairment (Mather et al., 2013). Previous studies have found that patients with rotator cuff tears exhibit more difficulty in the healing process of tendon-bone (Galatz et al., 2001). Currently, most clinical treatment strategies are based on surgery, in which the double-row fixation technique can improve rotator cuff repair and increase the likelihood of tendon-bone healing (Marrero, Nelman & Nottage, 2011; Dang & Davies, 2018). In recent years, cellular therapies, such as transplantation of allogeneic tendon-derived stem cells (TDSCs), have been explored for treating tendon injuries. TDSCs have universal stem cell characteristics like self-renewal, multidirectional differentiation, and clonogenicity (Bi et al., 2007). In rats, transplantation of allogeneic TDSCs has been found to inhibit the infiltration of T cells, mast cells, and macrophages and reduce inflammatory responses, thus promoting tendon repair (Lui et al., 2014).

In this research, we discovered different expression levels (DEGs) related to RCTs base on GEO datasets. Using the Degree algorithm and matching it with RCTs-related genes, we found 20 key target proteins connected to RCTs. Among these, we identified two central proteins, S100A1 and RASSF8. S100A1 belongs to the family of calcium-binding proteins involved in Ca2+ regulation in various tissues and organs. S100A1 is more frequently expressed in cardiac muscle, skeletal muscle fibers, and the brain (Donato et al., 2013). It reduces the interaction between S100A1 and eNOS and hyperactivation of protein kinase C in femur injury, thereby inhibiting eNOS phosphorylation, enhancing the degradation of vascular endothelial growth factor (VEGF) receptor-2 and diminishing VEGF signaling, resulting in impaired eNOS activity in endothelial cells, defective angiogenesis and increased amputation rates (Most et al., 2013). Besides its function in suppressing eNOS phosphorylation and enhancing angiogenesis, S100A1 also serves as a significant indicator in cellular therapies, particularly in cartilage regenerative cell therapy (Diaz-Romero et al., 2014). RASSF8 is a member of the Ras-associated structural domain family (RASSF) of tumor suppressor proteins. Overexpression of RASSF8 can inhibit cell growth and invasion, while reducing RASSF8 levels can increase NF-κB transcriptional activity and p65 translocation  (He et al., 2017). Furthermore, overexpression of RASSF8 can significantly inhibit cell apoptosis (Bo et al., 2018). In summary, both S100A1 and RASSF8 may promote the healing of rotator cuff tears by enhancing angiogenesis and inhibiting cell apoptosis.

This study found that the drug ZINC000026985532 (butanediamide) can target and bind to the S100A1 protein, forming a stable docking complex, based on the screening results of the FDA drug database. Butanediamide can inhibit the production of various inflammatory mediators and alleviate inflammatory reactions by suppressing the activation of multiple inflammatory signaling pathways. It also inhibits the activation of various inflammatory cells and promotes the activation of anti-inflammatory cells, thereby enhancing the body’s ability to fight inflammation (Alhonen et al., 2000; Miesel & Haas, 1993). In this study, we focused on the role of S100A1 in rotator cuff repair. S100A1 knockdown increased the proliferation of TDSCs, indicating that S100A1 can inhibit TDSC proliferation. We then explored the use of FDA-approved drug butanediamide for promoting rotator cuff repair. We found that butanediamide can promote the healing of rotator cuff tear by feeding 100 mg/kg butanediamide in a rat model of rotator cuff tears.

In conclusion, we have identified S100A1 and RASSF8 as potential therapeutic targets for rotator cuff tears. We also suggest that FDA-approved drug butanediamide may enhance rotator cuff healing by targeting S100A1 and RASSF8. Additional research is needed to determine the causal relationship between FDA-approved drug butanediamide and target genes.

Conclusions

In summary, this study offers new understanding into the possible use of FDA-approved drugs as treatment options for rotator cuff tears by focusing on the key proteins S100A1 and RASSF8. The results suggest that S100A1 has the ability to inhibit the growth of TDSCs. Moreover, the FDA-approved drug butanediamide may promote rotator cuff healing by inhibiting inflammatory responses and cell apoptosis.

Supplemental Information

Supplemental Information 1 Raw data of cck8

Click here for additional data file.

Supplemental Information 2 Uncropped GelsBlots of GAPDH and S100A1

Click here for additional data file.

Supplemental Information 3 Full (21-point) ARRIVE 2.0 checklist

Click here for additional data file.

Table S1 The top 20 compounds targered to S100A1 according to LibDockScore

Click here for additional data file.

Table S2 The top 20 compounds targered to RASSF8 according to LibDockScore

Click here for additional data file.

Table S3 Molecular docking results of the first six compounds targeting the core protein S100A1 binding

Click here for additional data file.

Table S4 Molecular docking results of the first six compounds targeting the core protein RASSF8 binding

Click here for additional data file.

Additional Information and Declarations

Competing Interests

Author Contributions

Animal Ethics

Data Availability

The authors declare there are no competing interests.

Yuan Lin conceived and designed the experiments, performed the experiments, analyzed the data, prepared figures and/or tables, and approved the final draft.

Ruipeng Guo conceived and designed the experiments, performed the experiments, analyzed the data, prepared figures and/or tables, and approved the final draft.

Geng R performed the experiments, analyzed the data, authored or reviewed drafts of the article, and approved the final draft.

Bin Xu conceived and designed the experiments, analyzed the data, authored or reviewed drafts of the article, and approved the final draft.

The following information was supplied relating to ethical approvals (i.e., approving body and any reference numbers):

All the experiments were approved by the Animal Research Ethics Committee of Southeast University (No. 20201231016).

The following information was supplied regarding data availability:

The transcriptome data of RCT are available at GenBank: GSE199484 and GSE93661.

https://www.ncbi.nlm.nih.gov/geo/query/acc.cgi?acc=GSE199484

https://www.ncbi.nlm.nih.gov/geo/query/acc.cgi?acc=GSE93661

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
