# Peer review of "Tailored modulation of S100A1 and RASSF8 expression by butanediamide augments healing of rotator cuff tears"

_PeerJ, doi:10.7717/peerj.15791_

## Round 0.1 · original submission · Major Revisions

· Academic Editor

Major Revisions

I completely agree with the opinions of the two reviewers. The manuscript is merely a bioanalysis, lacking in vivo and in vitro experimental verification, and the language needs to be polished. Based on the opinions of two reviewers, the decision of Major Revisions was made, hoping that the author would carefully modify the manuscript to address the concerns of reviewers.

Reviewer 1 ·

Basic reporting

Authors screened the common target proteins of both S100A1 and RASSF8 based on differentially expressed genes from rotator cuff tears, using a database of rotator cuff tears from various sources, and performed functional enrichment analyses of the target proteins using GO and KEGG. By further docking analysis by drug and target, Zinc compounds are able to elaborate the effects on rotator cuff tears in a very novel way. Further, this study provides an indication of the therapeutic mechanism of zinc compounds for the treatment of rotator cuff tears, and also gives a good idea for future research on the therapeutic mechanisms of compounds for trauma. However, there are some deficiencies in the writing of this paper, individual descriptions are not very accurate, and the study lacks complete in vivo and ex vivo experiments, which are not sufficient to support the reasoning.

Experimental design

*Venn diagrams for screening common target genes on two datasets are not clearly described in 1.2 of the experimental method.

*This study screens two different GEO databases for shared target genes. A brief explanation of why these two databases were selected is needed.

*The description language in experimental methods 1.3 and 1.4 is not reasonable.

Validity of the findings

*The description between two results can include transitional statements, such as by the above results, what should we study next? It is therefore necessary for the authors to describe which method was used and what results were obtained.

*Considering the results section is quite long, authors can summarize it, making it easier to read.

*Figure legends should be titled before the description of each figure follows.

*The figure legend of Figure 3 is too short, and S100A1 and RASSF8 can also be labeled on.

Additional comments

*Rather than Zinc compounds, I think it should be Butanediamide or another substance. It is recommended that the authors revise their manuscript for this and complete the results and discussion sections.

*It is also important that the manuscript's title be revised in this regard.

Reviewer 2 ·

Basic reporting

In this study, molecular biological mechanisms of rotator cuff tears were investigated and potential therapeutic agents were sought to promote rotator cuff repair. S100A1 and RASSF8 have been identified as the core differential proteins of RCT. S100A1 inhibits TDSC proliferation and exacerbates RCT symptoms. The authors also found that ZINC000049841054 affects S100A1 and ZINC000036701290 affects RASSF8 and participates in RCT repair.

Experimental design

The research questions in this manuscript are clear, relevant, and meaningful. However, the authors still need to explain how the study addresses this knowledge gap.

The manuscript does not adequately define the research question, as well. It is important for the authors to identify the knowledge gap under investigation and describe how the study contributes to filling it.

It is recommended that the authors add more references to methods in the text for the reader to learn, which will facilitate reproducibility of the experiment.

Validity of the findings

The results sections, including but not limited to the subsection "Virtual Screening of FDA Drug Database," should be streamlined and simplified.

It would be helpful if the authors could include some experiments to further demonstrate the potential role of the compounds; if not, please discuss this in the Discussion.

Additional comments

There is still a need to polish the manuscript with professional English language.

I think that it is not reasonable to mention the zinc compound in the background discussion because the authors wrote about it in the FDA drug database. It is therefore necessary to rewrite the background and discussion sections.

Discussions should have focused on the results and been more extensive.

In addition, "conclusions" should be stated appropriately, linked to the original question of the study, and limited to those that are supported by the results. Experimental interventions should support claims of causality in particular. It is recommended that the authors revise the manuscript in light of this suggestion.

---

## Round 0.2 · accepted · Accept

· Academic Editor

Accept

The authors have addressed the concerns of the reviewers.

Reviewer 1 ·

Basic reporting

no

Experimental design

no

Validity of the findings

no

Additional comments

I have carefully reviewed the revised manuscript titled "[Tailored Modulation of S100A1 and RASSF8 Expression by Butanediamide Augments Healing of Rotator Cuff Tears]" and I am pleased to recommend its acceptance for publication in the journal. The authors have adequately addressed all the concerns raised during the initial review process, and their revisions have substantially improved the quality and clarity of the manuscript.

Reviewer 2 ·

Basic reporting

no comment

Experimental design

no comment

Validity of the findings

no comment

Additional comments

The methodologies employed, which involved the creation of a protein-protein interaction (PPI) network and the subsequent identification of core genes, have been meticulous and methodologically sound. Moreover, the research has broadened our understanding of the biological roles and signal transduction pathways of these essential genes, thus propelling us closer to RCT-targeted treatment strategies.

The author's use of the Discovery Studio 2019 software for identifying potential FDA-approved drug targets, along with the subsequent in vivo and in vitro validation of drug efficacy, represents a robust and comprehensive approach to translational research. The results not only highlight S100A1 and RASSF8 as potential key players in RCTs, but also indicate that the FDA-approved compound, Butanediamide, could be a promising therapeutic agent for RCTs. I recommend this manuscript for acceptance and publication.